# Comparative Analysis of XGB, CNN, and ResNet Models for Predicting Moisture Content in *Porphyra yezoensis* Using Near-Infrared Spectroscopy

**DOI:** 10.3390/foods13193023

**Published:** 2024-09-24

**Authors:** Wenwen Zhang, Mingxuan Pan, Peng Wang, Jiao Xue, Xinghu Zhou, Wenke Sun, Yadong Hu, Zhaopeng Shen

**Affiliations:** 1Haide College, Ocean University of China, Qingdao 266003, China; wwz_mail_02@163.com; 2Jiangsu Coast Development Group Co., Ltd., Nanjing 210019, China; panmingxuan@jsyhkf.com (M.P.); yundiao2010@163.com (J.X.); zhouxinghu@jsyhkf.com (X.Z.); huyadong@jsyhkf.com (Y.H.); 3College of Food Science and Engineering, Ocean University of China, Qingdao 266003, China; pengwang@ouc.edu.cn (P.W.); 21220002081@stu.ouc.edu.cn (W.S.)

**Keywords:** *Porphyra yezoensis*, near-infrared (NIR) spectroscopy, moisture content, wavelength selection, prediction performance

## Abstract

This study explored the performance and reliability of three predictive models—extreme gradient boosting (XGB), convolutional neural network (CNN), and residual neural network (ResNet)—for determining the moisture content in *Porphyra yezoensis* using near-infrared (NIR) spectroscopy. We meticulously selected 380 samples from various sources to ensure a comprehensive dataset, which was then divided into training (300 samples) and test sets (80 samples). The models were evaluated based on prediction accuracy and stability, employing genetic algorithms (GA) and partial least squares (PLS) for wavelength selection to enhance the interpretability of feature extraction outcomes. The results demonstrated that the XGB model excelled with a determination coefficient (R^2^) of 0.979, a root mean square error of prediction (RMSEP) of 0.004, and a high ratio of performance to deviation (RPD) of 4.849, outperforming both CNN and ResNet models. A Gaussian process regression (GPR) was employed for uncertainty assessment, reinforcing the reliability of our models. Considering the XGB model’s high accuracy and stability, its implementation in industrial settings for quality assurance is recommended, particularly in the food industry where rapid and non-destructive moisture content analysis is essential. This approach facilitates a more efficient process for determining moisture content, thereby enhancing product quality and safety.

## 1. Introduction

*Porphyra yezoensis*, a representative species of edible seaweed, is primarily found in the western North Pacific Ocean and is commonly cultivated in China, Japan, and Korea [1,2]. Commercially known as “nori”, *P. yezoensis* products are sold in thin, dried, and seasoned sheets, known for their delightfully crispy texture and salty taste. Beyond its flavor, *P. yezoensis* boasts a range of distinctive proteins with antioxidant, anti-inflammatory, anticancer, and immunomodulatory properties. It also harbors numerous other bioactive compounds, such as carbohydrates, dietary fiber, and polyphenols, which offer promising health advantages and play crucial roles in food and agriculture [3].

The quality of dried *P. yezoensis* is dependent mainly on its moisture content. Insufficient drying can lead to corruption and degradation during transportation and storage due to excessive moisture content. Moreover, the highly hygroscopic nature of *P. yezoensis* can soften its crunchy texture, reducing customer satisfaction [4]. Therefore, it is necessary to develop an effective method for detecting moisture. Currently, many methods utilize the physical and chemical properties of food to determine moisture content, including Karl Fischer titration [5] and the direct drying technique [6]. Although these methods generally produce accurate results, they have drawbacks such as time-consuming procedures, unavoidable sample damage, and unsuitability for mass detection [7]. In the absence of efficient moisture detection techniques that do not compromise the integrity of the samples, the progress of *P. yezoensis* trade could be significantly impeded.

Near-infrared (NIR) spectroscopy is a moisture detection technology that has recently emerged to address the challenge of nondestructively inspecting food quality. It is widely used because of its rapid and high-volume processing characteristics. NIR provides spectral data in the wavelength range of 780–2500 nm, containing structural information about C-H, O-H, and N-H bonds [8]. The strongest NIR absorption bands in watery foods, such as sliced potato, pear flesh, and meat, appear at 1400–1410 nm [9]. However, relatively little research has been conducted on the specific range of absorption bands for processed dried foods, particularly dried seafood such as fish flesh, scallops, and *P. yezoensis*. Previous studies indicate that drying food can cause a nonlinear weakening and shift in the NIR absorption peak due to the reduction in moisture content. [10]. Furthermore, water absorption is highly sensitive to various factors, such as the presence of other ions and molecules, instrument instability, and external environmental factors [9,11]. These factors contribute to difficulties in extracting valid spectral information, resulting in low stability and prediction accuracy when using NIR to determine the moisture content of dried foods.

Currently, both traditional chemometric and deep learning algorithms are widely applied to analyze the NIR spectrum and efficiently predict the target chemical compound content [11,12]. Extreme gradient enhancement (XGB) is a popular chemometric algorithm that has been used in near-infrared analysis and has shown better predictive performance than other algorithms in many studies. Sousa et al. utilized four algorithms and near-infrared spectroscopy to classify the waxy phenotypes of cassava seeds, discovering that XGB outperformed the other three algorithms significantly [13]; the results from Yang et al. also demonstrated that XGB outperforms GBDT (gradient boosting decision tree) and RF (random forest) algorithms in predicting the moisture content of corn seeds [14]. According to Chen and Guestrin, XGB can perform up to ten times faster on a single machine than existing popular solutions and scale to even larger datasets using minimal cluster resources [15]. 

Additionally, more studies focus on the use of deep learning models, such as the convolutional neural network (CNN). Benmouna et al. built a quality detection model for Fujifilm Apple based on NIR and one-dimensional convolutional neural networks [16]; Zhang et al. combined near-infrared hyperspectral imaging and CNN for the determination of chemical compositions in dry black goji berries [17]. There have also already been numerous studies utilizing CNN algorithms and near-infrared spectroscopy to predict the moisture content of materials [18,19]. CNNs have fewer parameters than conventional neural networks, and their embedded regularization techniques enhance robustness against overfitting [20], greatly improving the model feature searching efficiency. Building on the architecture of CNNs, He et al. proposed a residual neural network (ResNet) in 2015 to address the degradation problem caused by gradient vanishing and explosions in CNNs [21]. It is generally used for various types of image analyses, such as medical images [22], garbage sorting [23], etc. Although researchers rarely employ ResNet in standard NIR analyses, its combination with Raman spectroscopy [24,25,26] and FT-NIR [27,28] already demonstrated its potential for spectroscopic analysis.

For fragile and perishable aquatic products, establishing rapid detection using near-infrared spectroscopy is highly beneficial. However, current research on NIR spectroscopy and various algorithms primarily focuses on staple foods and fruits, with fewer researchers directing their attention toward the quality inspection of aquatic products. Additionally, due to advanced prediction capabilities, deep learning models are often considered superior to chemometric algorithms for NIR analyses [17]. However, recent studies on tabular data highlighted the exceptional performance of XGB, noting its superiority over other deep learning models like 1D-CNNs [29]. Given that spectral data used in NIR studies differ structurally from the tabular data used in these comparisons, it is crucial to evaluate the effectiveness of XGB for processing NIR spectral data relative to other deep learning networks. 

This study aims to develop a non-destructive, rapid, and effective method for detecting moisture content that combines NIR spectroscopy with various predictive algorithms. This research encompasses the following objectives: (1) Employed GA and PLS to extract effective spectral information from dried food. (2) Compared the performance of the XGB, CNN, and ResNet models, determined the most accurate and stable prediction model for the moisture content of *P. yezoensis.* (3) Utilized Gaussian process regression (GPR) for a more comprehensive and stable model assessment.

## 2. Materials and Methods

### 2.1. Sample Preparation

Ideally, the dataset used to build an NIR calibration model should encompass as many sources of variation as possible. At a minimum, it must exhibit sufficient variability in geography, climate, processing methods, and growth stages. In this study, sample selection was meticulously planned to address these factors, aiming to meet the modeling requirements as comprehensively as possible. A total of 380 samples of *P. yezoensis* were collected from different sea areas (from origins such as Ganyu, Qinhuangdao, Qingdao, Weihai, etc.), processing enterprises, and harvesting periods (four batches of samples from November 2022 to May 2023), sourced from Laver Trading Co., LTD. (Ganyu, China). 

### 2.2. Moisture Content Determination

The moisture contents of the dried *P. yezoensis* samples were determined using a hydrotester (Model LHG20-A; Techcomp, Shanghai, China) that operates in accordance with the ISO 6540:2021 standard for the determination of moisture content in food [30]. This method involves drying the samples at 105 °C for 15–30 min until a constant weight is reached. The instrument uses an internal electronic scale to measure the weight difference and automatically calculate the moisture content.

### 2.3. Spectral Data Analysis

#### 2.3.1. Spectral Data Acquisition

The NIR spectral data were collected using a miniaturized handheld NIR spectrometer (VIAVI, Chandler, AZ, USA) with accompanying software. Spectral acquisition was conducted in a diffuse reflection-integrating sphere mode. The spectral range spanned from 900 to 1650 nm, with a data sampling interval of 3.0 nm, and each spectrum was averaged over 100 scans at 25 °C. Calibration involved using a 99% white diffuse reflectance standard, followed by dark measurements. Calibration was performed every 30 min. Before collecting samples, the device was preheated for 15 min, and the distance between the spectrometer head and the sample was maintained between 3 and 10 mm.

#### 2.3.2. Sample Set Split

The dataset, comprising 380 samples, was divided into a training set (300 samples) and a test set (80 samples). Several methods were proposed for dataset partitioning, including Kennard-Stone (KS), sample set partitioning based on joint X-Y distance (SPXY), and duplex [31]. The SPXY algorithm, which is an extension of the KS algorithm, takes into account both spectral variables (x) and chemical values (y) to create a more representative dataset [32]. In some NIR analysis cases, SPXY proved more effective than KS and other classification algorithms [31,33]. The training set was utilized to build the calibration models, while the test set was employed to evaluate the point and interval prediction performance of the optimized model.

#### 2.3.3. Spectral Pretreatment

When analyzing the collected raw spectral data, preprocessing methods were utilized to minimize or linearize the multiplicative and additive effects caused by light scattering during data collection [34] and to eliminate noise caused by the instrument and environment. Spectral preprocessing enhances the models’ ability to interpret spectral variations related to the concentration of compounds, thereby increasing accuracy and prediction capabilities. In this study, Savitzky–Golay (S-G) smoothing and the first derivative were employed simultaneously for noise reduction and baseline correction of the raw spectral data [35].

### 2.4. Wavelength Selection Method

Despite preprocessing, the full-spectrum data still contain a large number of variables that can increase model complexity and reduce training efficiency [36]. Therefore, to reduce data dimensions and enhance prediction effectiveness, it is necessary to eliminate non-informative wavelengths. In this study, the preprocessed data were analyzed using genetic algorithms (GA). GA draws on natural biological selection and genetic mechanisms, utilizing selection, crossover, and mutation operators. Through continuous genetic iterations, the variable with the best objective function value is retained, and the variable with the worst is eliminated, thus selecting the target characteristic variable [35]. The GA used in this study employs the negative mean squared error of the partial least squares (PLS) model as the fitness function, optimizing the feature subset by maximizing this function [37]. To assess the correlation between different wavelengths and the predicted moisture content, the regression coefficients of the PLS model were used to evaluate the importance of selected wavelengths.

### 2.5. Model Construction and Parameters Optimization

#### 2.5.1. XGB Model

Extreme gradient boosting (XGB) [15] is a scalable end-to-end tree-boosting system based on the gradient boosting decision tree framework [14]. It has been widely used to achieve State-of-the-Art results in various domains, such as store sales prediction, web text classification, customer behavior prediction, motion detection, product categorization, and hazard risk prediction. XGB is noted for performing ten times faster on a single machine and efficiently handling larger datasets with minimal cluster resources [15]. XGB introduces a regularized learning objective to optimize the learning process that enhances the model’s robustness, as illustrated in Equation (1).

(1)$$\begin{array}{c} obj=\sum_{i}l\left({\hat{y}}_{i},y_{i}\right)+\sum_{k}\Omega \left(f_{k}\right) \end{array}$$
where $\Omega \left(f\right)=\gamma T+\frac{1}{2}\lambda \;{∥\;\omega \;∥}^{\;\;2}$.

Here, l is a differentiable convex loss function that measures the difference between the prediction ${\hat{y}}_{i}$ and target $y_{i}$. The second term Ω penalizes the complexity of the model. Thus, the system eventually chooses the model with the least complexity and best predictive performance. Instead of optimizing the function using traditional methods, the model was trained using the additive algorithm shown in Equation (2).

(2)$$\begin{array}{c} obj=\sum_{i}^{n}l\left(y_{i},{{\hat{y}}_{i}}^{\left(t-1\right)}+f_{t}\left(x_{i}\right)\right)\;+\;\Omega \left(f_{t}\right) \end{array}$$
where the prediction of the i-th instance at the t-th iteration ${{\hat{y}}_{i}}^{\left(t\right)}$ is added to minimize the objective. Finally, a second-order approximation is utilized to rapidly optimize the objective and achieve an optimal solution.

#### 2.5.2. CNN Model

Convolutional neural networks (CNN) were first identified by Hubel and Wiesel in 1962 and introduced as a computational model in 1990 [38]. CNNs are multilayered, non-fully connected deep neural networks characterized by multiple repeated cycles of convolutional and pooling layers for feature extraction. Compared to traditional quantitative analysis models, CNNs offer advantages in automatic learning and parallel computation, which enhance generalization abilities and efficiency [39]. Additionally, CNNs can reduce the complexity of a model while maintaining efficient data processing capabilities compared to ordinary fully connected neural networks [39,40].

The CNN model used in this study is depicted in Figure 1d according to repeated training and debugging. The model consists of three one-dimensional convolutional layers (conv1d), two maximum pooling layers (maxpooling1d), and two fully connected layers (dense). Each input data point passes through the first convolution layer sequentially, with a size of 125 × 1, where each sample corresponds to the absorbance value at 125 specific spectral wavelengths. The first, second, and third convolution layers contain 64, 128, and 256 kernels, respectively. The two maximum pooling layers interspersed between these reduce the data dimensions. The output from the last pooling layer is flattened into a one-dimensional vector fed into the fully connected layers. These dense layers further reduce the data, enhancing the model’s expressive power through nonlinear transformations, eventually outputting the predicted moisture content. Other hyper-parameters for modeling, including learning rate and epoch, were fitted as 0.001 and 800, respectively. In this study, we held the condition that the epoch was used as the criterion for the end of training, rather than a satisfactory value of the loss function.

#### 2.5.3. ResNet Model

Previous studies indicate that depth in neural networks is crucial in improving model predictivity. However, adding more layers can exacerbate the problem of vanishing or exploding gradients. Although techniques like normalized initialization and intermediate normalization layers were proposed, increased depth can still degrade training accuracy. To address this, researchers developed a residual learning framework that employs a strategy known as residual mapping to facilitate training deeper networks without loss of performance [21], as illustrated in Figure 2.

In ResNet models, H(x) represents the desired mapping to be learned, where x is the output from the preceding layers. Rather than attempting to fit H(x) directly with stacked nonlinear layers, ResNet introduces a simpler approach by fitting a residual mapping F(x) = H(x) − x. This method simplifies optimization since optimizing the residual mapping F(x))) is generally more straightforward than optimizing the direct mapping (H(x) = F(x) + x. As depicted on the right side of Figure 2, ResNet incorporates a “shortcut connection” that performs identity mapping (x), and its output is added to the output of the stacked layers. Importantly, these shortcut connections do not increase model complexity, allowing the network to maintain robust performance while increasing the number of layers.

He et al. proposed several models with varying depths, including ResNet-18, ResNet-34, ResNet-50, ResNet-101, and ResNet-152 [21]. Given the limited size of our sample set and considering the distinct nature of NIR spectral data compared to the image data typically analyzed by ResNet, we opted for the ResNet-18 model in our experiment. We selected ResNet-18 due to its shallower architecture and modified it to better suit the unique characteristics of our one-dimensional spectral data. The architecture, as shown in Figure 1e, includes 16 convolutional layers within the four residual blocks, an additional convolutional layer and one max pooling layer at the beginning for initialization, and one average pooling layer. The learning rate and epoch were fitted as 0.0001 and 1200, respectively. 

#### 2.5.4. Gaussian Process Regression (GPR)

Predictive models exhibit a degree of uncertainty that impacts their performance when predicting unknown samples. To demonstrate the robustness and reliability of a model, it is essential to characterize its uncertainty level [41]. GPR is a robust probabilistic prediction algorithm well suited to complex regression challenges that are highly dimensional and nonlinear, and it can also facilitate interval prediction [42]. The GPR model can be described using formula X, where the input training set vector X, observed data Y, regression function f, and independent Gaussian white noise ε are integrated into the modeling process. This approach allows for a comprehensive uncertainty assessment, enhancing the model’s predictive reliability and robustness.

(3)$$\begin{array}{c} Y=f\left(x\right)+\varepsilon \end{array}$$
where $f\left(x\right)\;\sim \;N\left(\mu \left(X\right),k\left(X,X\right)\right);\;\varepsilon \;\sim \;N(0,\sigma_{n}^{2})$.

Then, the prior distribution of observation Y can be obtained.
(4)$$\begin{array}{c} Y\;\sim \;N\left(\mu \left(X\right),k\left(X,X\right)+\sigma_{n}^{2}I\right) \end{array}$$

Assuming the test set input vector X* and corresponding input Y*, the joint distribution of observation Y and test f(X*) is described as
(5)YfX* ~ NμXμX*,K+σn2IK*TK*K**
where I is a unit matrix and kernel functions ($K,K_{*},K_{**}$) represent the symmetric positive definite covariance matrix. In this study, Matern 5/2 was adopted as the kernel function [42]. Then, the posterior distribution and corresponding predicted value $\mu_{y}^{*}$ of the test set and covariance function $\sigma_{y}^{2*}$ can be obtained based on the Bayesian framework.
(6)$$\begin{array}{c} f\left(X^{*}|Y,X,X^{*}\right)\;\sim \;N\left(\mu^{*},\sigma^{2*}\right) \end{array}$$
(7)$$\begin{array}{c} \mu_{y}^{*}=K_{*}{\left(K+\sigma_{n}^{2}I\right)}^{-1}\left(Y-\mu \left(X\right)\right)+\mu \left(X^{*}\right) \end{array}$$
(8)$$\begin{array}{c} \sigma_{y}^{2*}=K_{**}-K_{*}{\left(K+\sigma_{n}^{2}I\right)}^{-1}K_{*}^{T}+\sigma_{n}^{2}I \end{array}$$

Finally, the 95% prediction interval can be calculated by [$\mu_{y}^{*}$ − 1.96$\sigma_{y}^{2*}$, $\mu_{y}^{*}$ + 1.96$\sigma_{y}^{2*}$].

The GPR algorithm was employed to calculate the prediction intervals to assess the uncertainties associated with the XGB, CNN, and ResNet models. The real values and prediction results from the training and test sets of these models were used as input data for the GPR algorithm, and then the corresponding $\mu_{y}^{*}$ and $\sigma_{y}^{2*}$ were obtained as the output to obtain the prediction interval of the model.

### 2.6. Statistical Analysis and Evaluation Metrics of the Models

#### 2.6.1. Evaluation Metrics of Point Prediction

The effectiveness of the models was evaluated using the coefficient of determination (R^2^), the ratio of performance to deviation for validation (RPD), and the root mean square error of prediction (RMSEP). Specifically, a model demonstrating a higher R^2^ and a lower RMSEP is considered to have a better fitting effect [12]. The RPD value is utilized for external verification, with a value above three indicating that the model is robust and suitable for screening applications [8]. The calculations for RPD, R^2^, and RMSEP are detailed in Equations (9)–(11), respectively. These metrics provide a quantitative basis for comparing the prediction accuracy and reliability of the different models under study.
(9)$$\begin{array}{c} RPD=\frac{1}{\sqrt{1-R^{2}}} \end{array}$$
(10)$$\begin{array}{c} R^{2}=1-\frac{\sum_{i=1}^{n}{\left(y_{i,actual}-y_{i,predicted}\right)}^{2}}{\sum_{i=1}^{n}{\left(y_{i,actual}-{\bar{y}}_{i,predicted}\right)}^{2}} \end{array}$$
(11)$$\begin{array}{c} RMSEP=\sqrt{\frac{\sum_{i=1}^{n}{\left(y_{i,actual}-y_{i,predicted}\right)}^{2}}{n-1}} \end{array}$$
where y_i,actual_ is the actual measured sample moisture value, y_i,predicted_ is the predicted sample moisture value, and n is the number of samples in the set. 

#### 2.6.2. Evaluation Metrics of Prediction Interval

In this study, we employed three specific metrics to assess the interval prediction error: prediction interval coverage probability (PICP), prediction interval normalized average width (PINAW), and coverage width criterion (CWC). PICP is used for uncertainty assessments by calculating the probability that observed values fall within the prediction intervals, with confidence levels typically ranging from 5% to 95% [41]. A higher PICP value signifies that more true values are encompassed within the predicted interval, indicating enhanced model stability. This metric is crucial for evaluating how well the model’s prediction intervals cover the range of actual outcomes, reflecting the reliability and confidence in the model’s predictive capabilities.
(12)$$\begin{array}{c} PICP=\frac{1}{N}\sum_{i=1}^{N}C_{i} \end{array}$$
$$C_{i}=\left\{\begin{array}{c} 1,\;L_{i}<y_{i}<U_{i}\\ 0,\;otherwise \end{array}\right.$$
where N is the number of samples in the dataset, and $C_{i}$ indicates the probability of observed value covered by prediction intervals. PINAW represents the average interval width of each observation point, which is important for testing whether the model is invalid because the larger the interval, the more likely it is that the observation will stay within the interval, i.e., the higher the PICP value [11].
(13)$$\begin{array}{c} PINAW=\frac{1}{NR}\sum_{i=1}^{N}\left(L_{i}-U_{i}\right) \end{array}$$

The interval prediction aims for high PICP and low PINAW [43]. Therefore, the CWC is defined, and its formula is as follows.
(14)$$\begin{array}{c} CWC=\frac{PICP}{PINAW} \end{array}$$

### 2.7. Software

The spectroscopic data were collected and exported using VIAVI MicroNIR Pro software. The data preprocessing, wavelength selection, and model construction were performed using Python version 3.10.13 as the programming language, with Visual Studio Code version 1.88 serving as the integrated development environment. The machine learning libraries employed were Scikit-learn version 1.3.2 and TensorFlow version 2.10.0. The figures were developed using MATLAB 2022b.

## 3. Results

### 3.1. Moisture Profile Analysis of Dried P. yezoensis

We divided the collected dried *P. yezoensis* samples into training and testing sets, as detailed in Table 1. This table illustrates that the means of the moisture levels in the test and training sets are similar, confirming a consistent moisture profile across the sets. The moisture content of the test set fell within the extremes observed in the training set, indicating rational sample partitioning. This is crucial for the validity of subsequent modeling and analytical processes.

### 3.2. Sample Moisture Profile and Spectrum Analysis Results

The unprocessed spectral data of 380 dried *P. yezoensis* samples are displayed in Figure 1a. After undergoing first-derivative pretreatment (Figure 1b), the irrelevant bandwidth was narrowed, and the spectrum became smooth without any burr noise. The absorption peak at 1180 nm was attributed to the C–H stretching second overtone, characteristic of the large number of carbohydrates in *P. yezoensis* [44]. Another absorption peak at 1430 nm, possibly related to the O–H stretching first overtone, was the strongest in the spectrum due to the dominant role of moisture in the near-infrared spectrum.

The GA selected 35 characteristic wavelengths ranging from 900 to 1650 nm, as depicted in Figure 3. The regression coefficients of the PLS model were utilized to ascertain the importance of the selected wavelengths. Higher coefficients corresponding to specific wavelengths indicate a greater contribution to the moisture content measurement. From Figure 3, it is evident that the wavelengths at 1392 and 1466 nm were most closely related to the prediction of *P. yezoensis* moisture content, likely due to the high polysaccharide content in dried *P. yezoensis* [45].

### 3.3. Moisture Content Prediction Results Using Three Different Models

#### 3.3.1. Point Prediction

Point prediction refers to the use of statistical or machine learning models to predict the value of a specific data point. According to Table 2, the XGB model, trained with full and selected spectra, demonstrated superior point prediction performance. The full-length XGB model achieved an R^2^ value of 0.979, an RPD value of 4.849, and an RMSEP value of 0.004, slightly outperforming the XGB model trained on the selected wavelength, with an R^2^ value of 0.975, an RPD value of 4.500, and an RMSEP value of 0.004. Figure 4a,b illustrates that the prediction points of the XGB model were closely aligned along the y = x line, indicating excellent model accuracy.

Based on the spectrum selected by the GA, the CNN model recorded an R^2^ value of 0.920, an RPD value of 2.546, and an RMSEP value of 0.008. The unselected ResNet model, with an R^2^ value of 0.900, an RPD value of 2.302, and an RMSEP value of 0.009, also effectively extracted spectral information for predicting the moisture content in dried *P. yezoensis*. The higher R^2^ and RPD values and lower RMSEP values indicated that the selected CNN model exhibited higher robustness and accuracy than ResNet.

Wavelength selection by the GA significantly enhanced the prediction performance of the CNN model, increasing the R^2^ value by 58.1%, the RPD by 108.5%, and reducing the RMSEP value by 137.5%. Conversely, for ResNet, the selected spectral information R^2^ led to a decrease in the value by 48.4%, the RPD by 51.0%, and an increase in the RMSEP value by 122.2%, which markedly reduced the model’s fitting performance and accuracy.

#### 3.3.2. Interval Prediction

Interval prediction provides a range of values within which future observations are expected to fall with a certain probability. It is a more comprehensive forecast that not only gives an estimated central value but also indicates the uncertainty around the point prediction. The XGB models based on the full and selected wavelengths exhibited the top two highest CWC values of 4.094 and 4. 041, respectively, indicating the highest reliability among all six models in Table 2. Additionally, Figure 5a–d depicts that the XGB models have the smallest shadow area, corresponding to the lowest PINAW value of 0.241.

The results for the CNN based on the selected band (CWC = 2.770) and ResNet (CWC = 2.486) also reflected low prediction uncertainty. Similar to point estimation, wavelength selection improved the performance of the CNN but worsened the performance of ResNet. Figure 5 and Table 2 show that the training models on the selected bands narrowed the shadow area of the CNN and reduced the PINAW by 66.1% but enlarged the shadow area of ResNet and increased the PINAW by 168.7%. It is noteworthy that even though the PICP values of the ResNet models both reached 1, the broad uncertainty shadow area (Figure 5c,f) with a PINAW up to 0.402 (based on full length) and 1.080 (based on selected wavelength) still indicates that the models’ uncertainty remains relatively high.

## 4. Discussion

Moisture content significantly influences chemical changes and microbial interactions in food, directly impacting its storage characteristics. For dried *P. yezoensis*, moisture affects both quality and the crispy texture, which are crucial for maintaining and controlling its quality [4,46]. Though accurate, traditional methods for detecting moisture content, such as the Karl Fischer method and direct drying method, are cumbersome and not well suited for large-scale detection or on-site production monitoring due to their long detection cycles and high energy consumption. In this study, we combined the strength of machine learning algorithms and near-infrared (NIR) spectroscopy to explore a fast, non-destructive water detection method for *P. yezoensis*. This study offered valuable insights for future research on moisture detection methods in similar dried seafood using NIR spectroscopy and contributes significantly to the theoretical foundation for quality control of *P. yezoensis*.

Using the GA, 35 effective wavelengths were successfully selected, enhancing the predictive performance of the CNN model. The regression coefficients from the PLS model highlighted two particularly critical wavelengths: 1391 nm and 1466 nm. The wavelength at 1466 nm was closely associated with the O-H stretching first overtone, and its strong correlation with moisture content was supported by numerous previous studies [9,47], affirming its importance in moisture detection in dried *P. yezoensis*. A wavelength of 1391 nm, although outside the typical water absorption peak range of 1400–1500 nm, demonstrated significant importance in predicting moisture in *P. yezoensis*. Research suggested that absorption at approximately 1360 and 1380 nm can be attributed to the combined effects of polysaccharides and water [45]. Therefore, it is reasonable to conclude that the uniquely high polysaccharide content influences the high coefficients of wavelengths around 1300–1400 nm in dried *P. yezoensis* [48,49]. For dried foods like *P. yezoensis* (also known as nori), the NIR spectrum is particularly sensitive to ion strength, gelatinization, swelling, and the presence of other ions and molecules, complicating the elimination of distractions and the capture of useful spectral information [9].

The outstanding prediction performance of the XGB models proposed in this study demonstrated their suitability for analyzing moisture content based on complex and nonlinear spectral data. The full-length XGB model slightly outperformed the XGB model using selected wavelengths, indicating better feature selection capability than the GA. Similar previous studies in different domains, such as the prediction and classification of grape maturity [50], crop intensity mapping [51], and waxy phenotype classification in cassava seeds [13], also highlighted the strong performance of XGB. The results aligned with a report [29], which compared XGB with four other deep learning models, including a 1D-CNN, finding that XGB exhibited a 50% lower average relative performance (a lower value is better) than the deep learning models while converging in a shorter runtime. Overall, the XGB model has significant advantages in terms of prediction accuracy and processing speed and can monitor moisture content in real time in large-scale production sites. In addition, the high stability and low prediction error of the XGB model enables it to maintain a consistent prediction performance over the uncertainties of the instrument itself and the errors brought about by complex sample sources such as origin, management mode, and planting years, which is essential for maintaining consistent product quality [11].

While the CNN and ResNet models have the potential for better performance, they require significantly more time and resources to optimize hyperparameters in real-world applications than XGB. Deep learning models, often described as a “black box,” lack transparency or interpretability in transforming input data into outputs [52]. They involve a large number of nonlinear transformations and complex network structures. This may require more professional abilities and larger sample sets to achieve the desired prediction performance. At the same time, in food quality testing, if the model does not provide a clear basis for decision making, it may be difficult to gain the trust of regulators and consumers even if the prediction is accurate. In contrast, the XGB model provides better interpretability. The tree structure of the XGB model makes its decision-making process more transparent. This interpretability not only helps in the debugging and optimization of the model but also enables the end user to understand the basis of the prediction of the model, thus enhancing the credibility and acceptance of the model.

Although ResNet did not emerge as the best performing model in this study, the results highlighted its potential for NIR analysis. Originally, ResNet was designed to process two-dimensional or three-dimensional data, such as images [21]. However, NIR spectroscopy involves one-dimensional data, suggesting that modifications to the original ResNet architecture might be necessary to enhance its compatibility with NIR spectra. ResNet’s ability to continuously add depth without complicating the model [21] enables superior feature extraction and data processing capabilities. It can detect subtle spectral changes that many other algorithms might miss [26]. This sensitivity could explain why ResNet, based on selected spectral data, underperformed in this context—the GA-selected spectra likely omitted information deemed critical by ResNet.

Previous research showed that ensemble models can significantly enhance prediction performance and reduce variance by leveraging the strengths of submodels [29]. Various ensemble models using XGB as the base learner were successfully implemented [14,29]. Future efforts could explore an ensemble approach incorporating XGB, ResNet, or other deep learning models to achieve a more robust prediction performance for dried *P. yezoensis*.

## 5. Conclusions

In this study, the XGB model based on the full spectrum demonstrated a superior prediction performance than the CNN and the ResNet models. GA and PLS algorithms successfully identified 35 effective wavelengths, which were significantly correlated with the moisture content of dried *P. yezoensis*. In actual production sites, manufacturers can install a near-infrared spectrometer equipped with the model developed in this study on the production line at the end of the drying process for *P. yezoensis*. This installation allows for real-time moisture monitoring of the dried nori samples, thereby achieving efficient control and management of the nori quality.

While this study has yielded satisfactory outcomes, expanding the wavelength range could further enhance the applicability of our findings. The wavelength range measured in this experiment was relatively narrow, hence bands that may be relevant to this experiment, such as the free OH combination tone band (1920–1980 nm), were not included in the datasets. Future research could focus on designing ensemble models that combine the strengths of XGB with other deep learning models to further refine the analysis of the NIR spectrum and enhance the accuracy of moisture content predictions for dried *P. yezoensis*.

## Figures and Tables

**Figure 1 foods-13-03023-f001:**
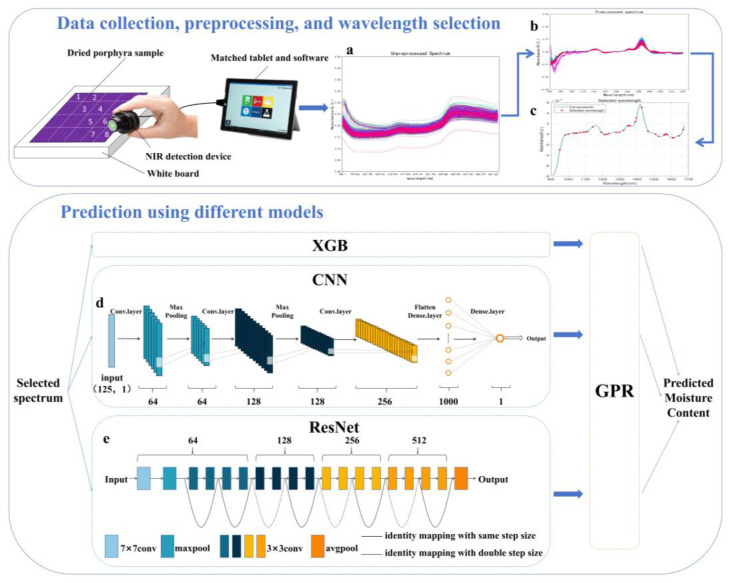
The collection, preprocessing, and wavelength selection process of *cfv* and prediction process using XGB, CNN, and ResNet methods. (**a**): Raw spectral data of the *P. yezoensis* samples; (**b**): spectral data after pretreatment; (**c**): wavelength selection result; (**d**): the architecture of CNN; and (**e**): the architecture of ResNet.

**Figure 2 foods-13-03023-f002:**
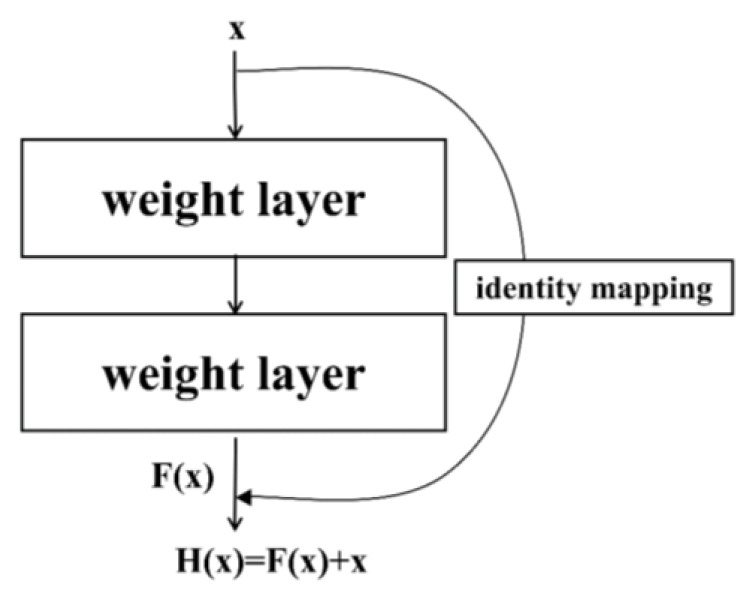
Residual block structure in ResNet.

**Figure 3 foods-13-03023-f003:**
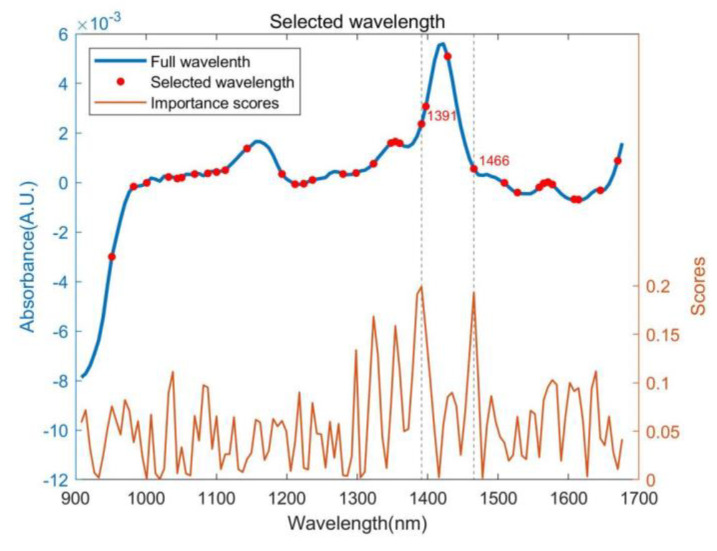
Wavelengths selected by GA.

**Figure 4 foods-13-03023-f004:**
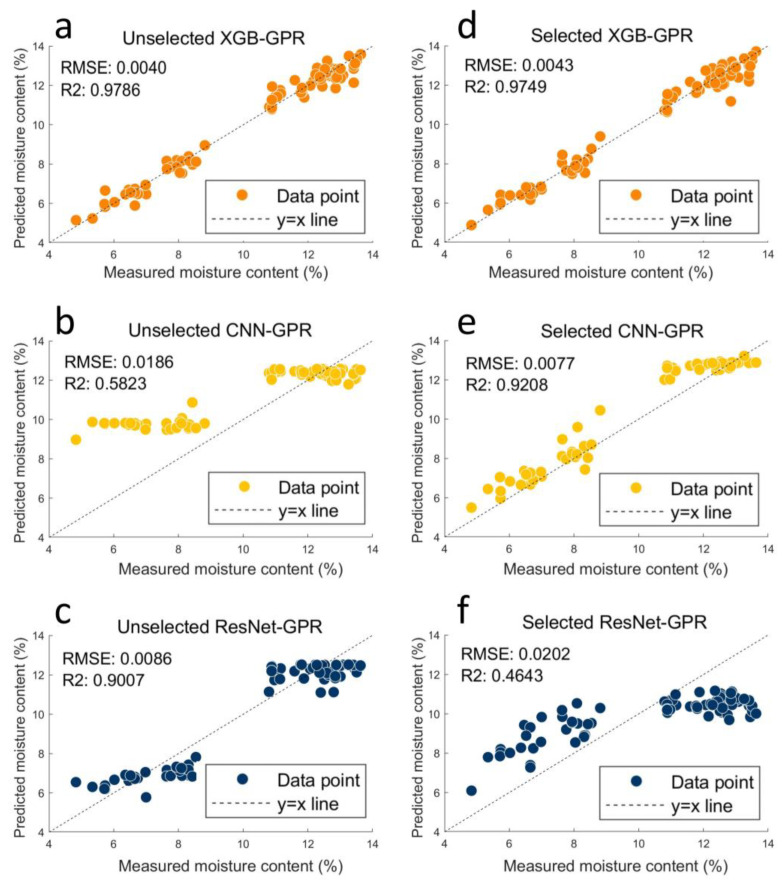
The prediction results of XGB, CNN, and ResNet based on full wavelengths and selected wavelengths: (**a**) point prediction results of XGB based on full wavelengths; (**b**) point prediction results of CNN based on full wavelengths; (**c**) point prediction results of ResNet based on full wavelengths; (**d**) point prediction results of XGB based on selected wavelengths; (**e**) point prediction results of CNN based on selected wavelengths; and (**f**) point prediction results of ResNet based on selected wavelengths.

**Figure 5 foods-13-03023-f005:**
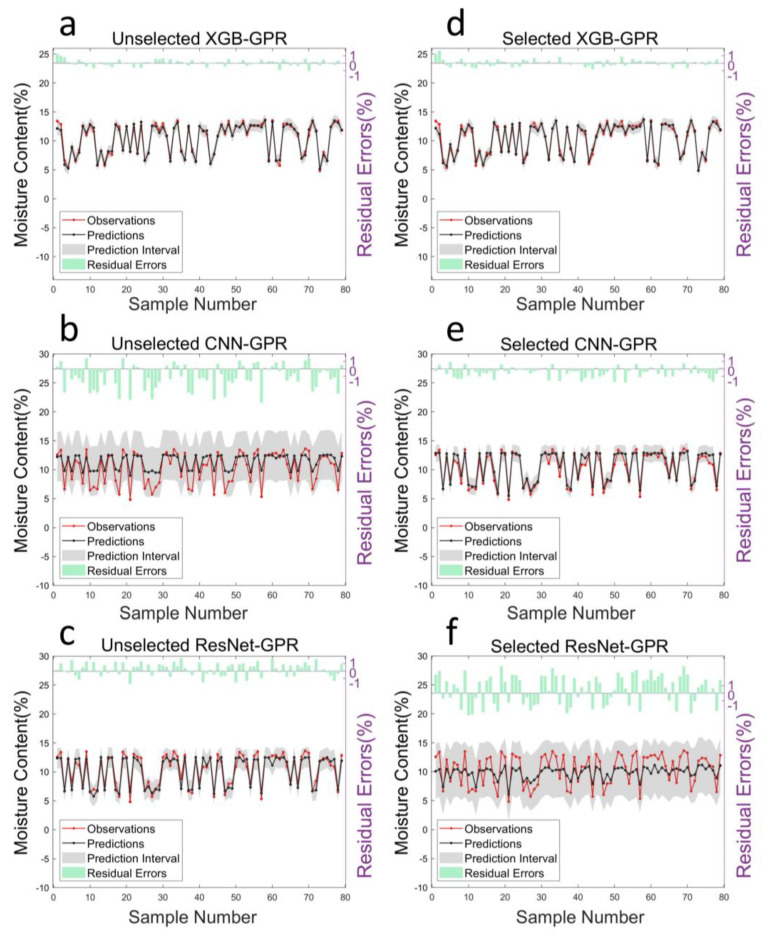
The prediction intervals and residual errors of XGB, CNN, and ResNet based on full wavelengths and selected wavelengths: (**a**) prediction intervals and residual errors of XGB based on full wavelengths; (**b**) prediction intervals and residual errors of CNN based on full wavelengths; (**c**) prediction intervals and residual errors of ResNet based on full wavelengths; (**d**) prediction intervals and residual errors of XGB based on selected wavelengths; (**e**) prediction intervals and residual errors of CNN based on selected wavelengths; and (**f**) prediction intervals and residual errors of ResNet based on selected wavelengths.

**Table 1 foods-13-03023-t001:** The descriptive statics of the training set and test set.

Dataset	Moisture Statics					
	Number of Samples	Average Value (%)	Maximum Value (%)	Minimum Value (%)	Medium Value (%)	Standard Deviation (%)
Training set	300	10.18	14.84	4.37	7.91	2.78
Test set	80	9.93	14.05	4.45	7.82	1.24

**Table 2 foods-13-03023-t002:** The point prediction results (the values of R^2^, RMSEP, and RPD) and interval prediction results (the values of PICP, PINAW, and CWC) using XGB, CNN, and ResNet models based on full wavelengths and selected wavelengths.

Wavelength Selection	Models	Point Prediction			Interval Prediction		
		R^2^	RMSEP	RPD	PICP	PINAW	CWC
Full wavelength	XGB	0.979	0.004	4.849	0.987	0.241	4.094
Full wavelength	CNN	0.582	0.019	1.230	0.987	0.970	1.017
Full wavelength	ResNet	0.900	0.009	2.302	1.000	0.402	2.486
Wavelength selected by Genetic Algorithm	XGB	0.975	0.004	4.500	0.975	0.241	4.041
Wavelength selected by Genetic Algorithm	CNN	0.920	0.008	2.564	0.911	0.329	2.770
Wavelength selected by Genetic Algorithm	ResNet	0.464	0.020	1.129	1.000	1.080	0.926

R^2^, the coefficient of determination; RMSEP, the root mean square error of prediction; RPD, the ratio of performance to deviation for validation; PICP, the prediction interval coverage probability; PINAW, prediction interval normalized average width; and CWC, coverage width criterion.

## Data Availability

The original contributions presented in this study are included in the article, and further inquiries can be directed to the corresponding author.

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
