# Peer review of "Comparative Analysis of XGB, CNN, and ResNet Models for Predicting Moisture Content in Porphyra yezoensis Using Near-Infrared Spectroscopy"

_foods, 2024, doi:10.3390/foods13193023_

Round 1
Reviewer 1 Report
Comments and Suggestions for Authors
1. The paper does a good job analyzing predictive models for moisture content in Porphyra yezoensis using NIR spectroscopy, but it lacks a dedicated related work/literature review section. Including this section is essential for positioning the study within the broader context of existing research, framing the research questions, comparing it with previous studies, identifying gaps, and emphasizing the novelty of the work, particularly in the application of XGB, CNN, and ResNet models.
2. The discussion could be improved by linking the findings more directly to practical applications. For example, how does the superior performance of the XGB model enhance non-destructive moisture detection in an industrial setting? You can add a practical implications subsection in the Discussion to highlight this.
3. The paper touches on the complexity of deep learning models, but it might benefit from further exploration of the trade-off between performance and interpretability. How might the “black box” nature of CNN and ResNet models affect their use in practical applications compared to the more interpretable XGB?
4. The methodology section could be improved by discussing the model training process more critically. Providing more information on how hyperparameters were selected and tuned for each model would help readers better understand the effort involved.
5. The text in Figures 4 and 5 is too small and difficult to read. Please improve them.
Comments on the Quality of English Language
1. The paper could benefit from consistent use of tense, particularly in the abstract and discussion sections, where shifting between past and present tense occurs.
2. The paper predominantly uses passive voice, which can sometimes make the text less direct. Revising some sentences to active voice could make the writing more engaging and clearer for the reader.
Author Response
Comment 1.The paper does a good job analyzing predictive models for moisture content in Porphyra yezoensis using NIR spectroscopy, but it lacks a dedicated related work/literature review section. Including this section is essential for positioning the study within the broader context of existing research, framing the research questions, comparing it with previous studies, identifying gaps, and emphasizing the novelty of the work, particularly in the application of XGB, CNN, and ResNet models.
We sincerely appreciate the valuable comments. We have checked the literature carefully and added more reference on the application of XGB and CNN, at line 68-77 and line 79-84 respectively. Meanwhile, we added some more illustrations at line 93-96 to point out the insufficiency between the current researches.
The new references are as follows:
- Sousa, M.B. e; Filho, J.S.S.; de Andrade, L.R.B.; de Oliveira, E.J. Near-Infrared Spectroscopy for Early Selection of Waxy Cassava Clones via Seed Analysis. Front. Plant Sci. 2023, 14, doi:10.3389/fpls.2023.1089759.
- Yang Lin; Zhang Lin; Ye Zehui Content in Maize Seeds Based on Ensemble Learning and Near Infrared Spectroscopy. Acat Agriculturae Boreali-Occidentalis Sinica 2022, 31, 1025–1034.
- Benmouna, B.; García-Mateos, G.; Sabzi, S.; Fernandez-Beltran, R.; Parras-Burgos, D.; Molina-Martínez, J.M. Convolutional Neural Networks for Estimating the Ripening State of Fuji Apples Using Visible and Near-Infrared Spectroscopy. Food Bioprocess Technol 2022, 15, 2226–2236, doi:10.1007/s11947-022-02880-7.
- Zhang, C.; Wu, W.; Zhou, L.; Cheng, H.; Ye, X.; He, Y. Developing Deep Learning Based Regression Approaches for Determination of Chemical Compositions in Dry Black Goji Berries (Lycium Ruthenicum Murr.) Using near-Infrared Hyperspectral Imaging. Food Chemistry 2020, 319, 126536, doi:10.1016/j.foodchem.2020.126536.
- Cui, J.; Sawut, M.; Ailijiang, N.; Manlike, A.; Hu, X. Estimation of Leaf Water Content of a Fruit Tree by In Situ Vis-NIR Spectroscopy Using Multiple Machine Learning Methods in Southern Xinjiang, China. Agronomy 2024, 14, 1664, doi:10.3390/agronomy14081664.
- Liu, Y.; Lu, Y.; Chen, D.; Zheng, W.; Ma, Y.; Pan, X. Simultaneous Estimation of Multiple Soil Properties under Moist Conditions Using Fractional-Order Derivative of Vis-NIR Spectra and Deep Learning. Geoderma 2023, 438, 116653, doi:10.1016/j.geoderma.2023.116653.
Comment 2.The discussion could be improved by linking the findings more directly to practical applications. For example, how does the superior performance of the XGB model enhance non-destructive moisture detection in an industrial setting? You can add a practical implications subsection in the Discussion to highlight this.
Thank you for your suggestions! At line 458-464 we carefully illustrate how the prediction accuracy and processing speed of XGB can be beneficial during actual industrial setting.
Comment 3. The paper touches on the complexity of deep learning models, but it might benefit from further exploration of the trade-off between performance and interpretability. How might the “black box” nature of CNN and ResNet models affect their use in practical applications compared to the more interpretable XGB?
Thanks for your advice, we are indeed too vague about black box effects for deep learning models. According to your suggestions, we explained what the “black box” actually mean and its impact to the engineers and consumers at the food industry at line 468-473. We also explained how XGB model can avoide “black box” situations and be superior than CNN and ResNet at line 473-477.
Comment 4. The methodology section could be improved by discussing the model training process more critically. Providing more information on how hyperparameters were selected and tuned for each model would help readers better understand the effort involved.
Thank you for your advice. To help readers better understand how we achieved the hyperparameters of the models, we added some illustrations at line 209-210, line 219-222, line 242-247 and line 250.
Comment 5.The text in Figures 4 and 5 is too small and difficult to read. Please improve them.
We have rearrange the layout of figure 4 and 5, and larger the font size in order to make it more readable. To ensure that the figures are not compressed during the insertion of text, we also provide PDF version of the figures.
Comment 6. The paper could benefit from consistent use of tense, particularly in the abstract and discussion sections, where shifting between past and present tense occurs.
We think this is an excellent suggestion. According to your advice, we have change all the tense at the abstract section to past tense and tried to unify the verb tenses in the discussion part into the present tense, excluding some statements that better use the past tense to describe previous researches.
Comment 7. The paper predominantly uses passive voice, which can sometimes make the text less direct. Revising some sentences to active voice could make the writing more engaging and clearer for the reader.
We strongly agree with your suggestion and have made the following modifications:
Line 58-60, “Previous studies indicate that the NIR absorption peak in dried food can be weakened and shifted nonlinearly due to decreased moisture.” → “Previous studies indicate that drying food can cause a nonlinear weakening and shift in the NIR absorption peak due to the reduction of moisture content.”
Line 87, “a Residual Neural Network (ResNet) was proposed in 2015 ” → “He et al. proposed a Residual Neural Network (ResNet) in 2015”
Line 90, “Although ResNet is rarely used in standard NIR analyses ” → “Although researchers rarely employ ResNet in standard NIR analyses”
Line 301, “In this study, three specific metrics were employed to assess the interval prediction error……” → “In this study, we employed three specific metrics to assess the interval prediction error……”
Line 331, “The dried P. yezoensis samples that were collected were divided into training and testing sets……” → “We divided the collected dried P. yezoensis samples into training and testing sets”

Reviewer 2 Report
Comments and Suggestions for Authors
Dear authors,
The MS entitled „Comparative Analysis of XGB, CNN, and ResNet Models for Predicting Moisture Content in Porphyra yezoensis Using Near-Infrared Spectroscopy“ by Zhang et al., has been reviewed. In the present work, author ivestigated performance of the XGB, CNN, and ResNet models for predicting moisture content in P. yezoensis using NIR spectroscopy. Research results showed that some wavelengths significantly correlate with moisture content of dried P. yezoensis. Some specific comments are given below.
Comments
1. Line 14: „porphyra yezoensis“ should be „Porphyra yezoensis“
2. Line 80: Please delete the bracket XGB)
3. Lines 98-102: It is not clear sample selection and preparation, please add more detail information about samples and place where are samples colleted.
4. Lines 104-106: Please add more details about determine moisture content (according to method reference), and for example how long samples were exposed
5. Figure 1., Figure 4. and Figure 5, should be better presented. Letters should be visible (same latter size), please improve resolution.
6. Lines 301:” 3.1. moisture profile analysis of dried P. yezoensis” change to “3.1. Moisture profile analysis of dried P. yezoensis”
7. Lines 309, 313, 323, 324, P. yezoensis – it should be italic
Author Response
- Line 14: “porphyra yezoensis“ should be“Porphyra yezoensis”
Thank you for your careful reading, we have corrected this error at line 14.
- Line 80: Please delete the bracket XGB)
Thank you for your careful reading, we have corrected this error at line 99.
- Lines 98-102: It is not clear sample selection and preparation, please add more detail information about samples and place where are samples colleted.
Thank you for your suggestions, we have added some extra information on the origin and harvest period of the samples at section 2.1.
- Lines 104-106: Please add more details about determine moisture content (according to method reference), and for example how long samples were exposed
Thank you for your valuable suggestions regarding the moisture content determination section of our manuscript. We appreciate your guidance on ensuring a more detailed illustrations of our method.
In response to your recommendation, we have updated the section to include a reference to the ISO 6540:2021 standard, which our method aligns with and added the temperature and time that required for our method.
- Figure 1., Figure 4. and Figure 5, should be better presented. Letters should be visible (same latter size), please improve resolution.
We have rearrange the layout of figure 4 and 5, and larger the font size in figure 1,4,5 to make it more readable. To ensure that the figures are not compressed during the insertion of text, we also provide PDF version of the figures.
- Lines 301:” 3.1. moisture profile analysis of dried P. yezoensis” change to “3.1. Moisture profile analysis of dried P. yezoensis”
Thank you for your careful reading , we have corrected this error.
- Lines 309, 313, 323, 324, P. yezoensis – it should be italic
We are sorry for our carelessness, and have corrected all the “P. yezoensis” to italic.

Reviewer 3 Report
Comments and Suggestions for Authors
Lines 8-21 The abstract could be improved adding some more experimental details, some summarizing results and a short operative suggestion
L. 28 Change “uncertainty assessment” within the keywords
Lines 55-56. Too much generic sentence…what kind of dried foods did you refere?
L. 85-93. Shorten the aim and be more explicative
Lines 100-102 Add more details about the sampling dataset… season of collection, marine sites, etc
Line 104-105. Is according to an AOAC methodology? Add more details and all over report a connection with an official wet chemistry methods in order to determine the moisture content.
L. 256-257 and 261, Do you mean R2 (why Rp2?)
L. 301 and 309, P. yezoensis
L. 307. Descriptive statistics
L. 307. Table 1. Add the standar deviation
L. 317. Reported Genetic algorithm instead GA (figure should be fully readness)
L. 326. “point prediction results” …it’s not very clear, improve
L. 326 Table 2. Reported in a end-table caption the meaning of R2, RMSEP, RPD
L. 326 Table 2. Not clear the first column…
L. 328 Not clear “Point prediction”
L. 330 and 332 and son on: R2
L. 379-382. Be carefull. This is part of the template …
L. 383-388. It sounds like introduction. Move on the introduction or delete
L. 423-425 Not clear. Explain better
L. 452-454. Avoid to use results in the conclusions
L. 450 Conclusions. Improve. Avoid to repeat the results. Add an operative comment
Comments on the Quality of English LanguageEnglish language and style should be improved. There are many minor grammar and editing mistakes.
Author Response
- Lines 8-21 The abstract could be improved adding some more experimental details, some summarizing results and a short operative suggestion
Thank you for your suggestions! We have rewritten the abstract section. We added some more experimental details at line 15-17 and 22-23, and a short operative suggestion at the end of the abstract.
- Line 28 Change “uncertainty assessment” within the keywords
The keyword “uncertainty assessment” has been changed to “prediction performance”
- Lines 55-56. Too much generic sentence…what kind of dried foods did you refer?
Sorry about the vague description. We have changed the sentences at line 56 and line 58 to more specific illustrations.
- 85-93. Shorten the aim and be more explicative
Thank you for your suggestions, we have rewritten the last paragraph of the introduction to make it more explicative and more brief.
- Lines 100-102 Add more details about the sampling dataset… season of collection, marine sites, etc
Thank you for your suggestions, we have added some extra information on the origin and harvest period of the samples at section 2.1.
- Line 104-105. Is according to an AOAC methodology? Add more details and all over report a connection with an official wet chemistry methods in order to determine the moisture content.
Thank you for your valuable suggestions regarding the moisture content determination section of our manuscript. We appreciate your guidance on ensuring a connection with an official wet chemistry method.
In response to your recommendation, we have updated the section to include a reference to the ISO 6540:2021 standard, which our method aligns with. The moisture content determination process we employed is based on the principle of drying the sample at an elevated temperature until a constant weight is achieved, allowing for the calculation of moisture content through the weight difference.
- 256-257 and 261, Do you mean R2 (why Rp2?)
Sorry about our mistakes, we have changed all “Rp2” to R2.
- 301 and 309, P. yezoensis
We have corrected this error.
- 307. Descriptive statistics
According to your suggestion, we have added standard deviation and medium value to enrich our analysis of the descriptive data of the sample dataset.
- 307. Table 1. Add the standard deviation
Thank you for your suggestions, we have added the standard deviation of the training and test set.
- 317. Reported Genetic algorithm instead GA (figure should be fully readness)
We have change GA to “Genetic algorithm” in table 2.
- 326. “point prediction results” …it’s not very clear, improve
We have added the the description of the three parameters referred to by the specific point prediction results. In order for the reader to better understand the meaning of the text, we added the definition of point prediction and interval prediction on line 360-361 and line 386-389.
- 326 Table 2. Reported in a end-table caption the meaning of R2, RMSEP, RPD
Thank you for your advice! We have added a end-table caption the meaning of R2, RMSEP, RPD, PICP, PINAW and CWC in Table 2.
- Line 326 Table 2. Not clear the first column…
We have changed our illustrations at the first column of table 2.
- 328 Not clear “Point prediction”
I am very sorry that we do not have the relevant explanation here. "Point prediction" here refers to the use of statistical or machine learning models to predict the value of a specific data point. We have added the relevant explanation in the first paragraph of 3.3.1 in the article.
- 330 and 332 and son on: R2
Sorry about the editing errors, we have changed all the R2 in the article to R2.
- 379-382. Be carefull. This is part of the template …
We have deleted line 412-415.
- 383-388. It sounds like introduction. Move on the introduction or delete
We have deleted the corresponding sentences at line 424-429.
- 423-425 Not clear. Explain better
Thanks for your advice, we are indeed too vague about black box effects for deep learning models. According to your suggestions, we explained what the “black box” actually mean and its impact to the engineers and consumers at the food industry. We also explained how XGB model can avoide “black box” situations and be superior than CNN and ResNet. (Line 468-477)
- 452-454. Avoid to use results in the conclusions
Same as suggestion 21.
- Line 450 Conclusions. Improve. Avoid to repeat the results. Add an operative comment
Thank you for your suggestions. We have rewritten the conclusion, deleted the repetition of the results and added some shortcomings of this study and operative comments.
22.English language and style should be improved. There are many minor grammar and editing mistakes.
Thanks for your suggestions! We have gone over the article from beginning to end to correct the language, grammar and editing errors in our articles:
Line 14“porphyra yezoensis“ to“Porphyra yezoensis”
Line 78 “focuses” to “focus”;
Line 78 “Convolutional” to “the Convolutional”;
Line 99 “XGB)” to “XGB”;
Corrected all the “P. yezoensis” to italic.
Unified the verb tenses in the discussion part into the present tense, excluding some statements that better use the past tense to describe previous researches.

Round 2
Reviewer 1 Report
Comments and Suggestions for Authors
The authors have made substantial revisions to the manuscript and addressed all my comments properly. Therefore, the paper is suitable for acceptance in its current form.